# Functional human genes typically exhibit epigenetic conservation

Daniel Rud[1], Paul Marjoram[1], Kimberly Siegmund[1], Darryl Shibata[2]*

1 Department of Preventive Medicine, University of Southern California, Keck School of Medicine, Los Angeles, CA, United States of America, 2 Department of Pathology, University of Southern California, Keck School of Medicine, Los Angeles, CA, United States of America

* dshibata@usc.edu

## Abstract

Recent DepMap CRISPR-Cas9 single gene disruptions have identified genes more essential to proliferation in tissue culture. It would be valuable to translate these finding with measurements more practical for human tissues. Here we show that DepMap essential genes and other literature curated functional genes exhibit cell-specific preferential epigenetic conservation when DNA methylation measurements are compared between replicate cell lines and between intestinal crypts from the same individual. Culture experiments indicate that epigenetic drift accumulates through time with smaller differences in more functional genes. In NCI-60 cell lines, greater targeted gene conservation correlated with greater drug sensitivity. These studies indicate that two measurements separated in time allow normal or neoplastic cells to signal through conservation which human genes are more essential to their survival in vitro or in vivo.

## Introduction

Only subsets of genes are expressed in different cell types. It would be valuable to determine which genes are more critical to the survival of specific human cell types. A gene may be considered essential when loss of its function compromises the viability or fitness of its cell [1, 2]. A rigorous approach to determine essentiality is to inactivate a gene and determine subsequent viability. Recent CRISPR-Cas9 studies performed single gene disruptions of cancer cell lines, identifying hundreds of essential human genes [1–3]. However, similar in vivo experimental approaches in human tissues are currently unethical, and translation of cell culture findings is problematic because essentiality is cell type and context dependent [1, 2].

Here we show that epigenetic conservation can serve as an orthogonal noninterventional metric that can infer gene function in human tissues. Epigenomes are readily measured, and quantifying conservation simply requires at least two measurements separated by sufficient time to allow for drift to accumulate in less essential genes. In general, functional regions of a genome are more conserved because random alterations are often deleterious, leading to reduced fitness and loss due to negative or purifying selection [4]. Potentially somatic cell epigenomes are also subject to negative selection because robust mammalian gene expression depends on its epigenetic configuration [5]. Drift in non-essential genes is more likely to be

**Data Availability Statement:** The data underlying this study are available on the GEO database (GSE182772).

**Funding:** This work was supported by NIH Grant U54 U54CA217376 (DS) The funders had no role

in study design, data collection and analysis, decision to publish, or preparation of the manuscript.

**Competing interests:** The authors have declared that no competing interests exist.

tolerated, whereas changes in functional genes may alter an optimal epigenetic configuration, leading to decreased cell fitness, lower proliferation, and cell loss. Therefore, epigenetic conservation could be used as a general unbiased noninterventional metric to identify functional genomic regions in somatic cells.

We have previously used DNA methylation to illustrate that gene expression correlates with epigenetic conservation [6], and that genes involved in immune surveillance are preferentially conserved during human colorectal cancer growth [7]. The DepMap single gene CRISPR-Cas9 disruption data [3] provide genome-wide tests of whether epigenetic conservation correlates with gene function. We find that the epigenetic configurations of DepMap and other important genes are preferentially conserved during tissue culture and in human tissues, allowing inferences of gene function without prior experimental manipulations.

## Results

### How the method works

Here we outline how epigenetic conservation can correlate with gene function in somatic cells. DNA methylation is measured with arrays, which have high precision that minimizes technical variations [8]. The commercial arrays and standardized bioinformatics [9] facilitate comparisons between samples and experiments. Two measurements of the same sample are required, and the method records what happens to an epigenome between these observations. The epigenetic configuration of a gene is subject to selection because it controls its expression [5]. Selection requires variation and therefore time, because two daughter cells initially share nearly identical epigenomes (Fig 1A). Random replication errors or drift will accumulate with time, and epigenomes will progressively become different. These differences are measured as the absolute differences in average DNA methylation (beta values) at a CpG site, or a pairwise difference (PWD) that can range from 0 to 1. Changes that accumulate in non-genic regions or in genes without critical functions in a cell type likely have no survival consequences, but changes in functional genes may alter cell fitness.

Initially it may be difficult to use conservation to infer gene function because differences may confer both positive and negative changes. However, in most cases, human cells are sampled many years after they shared a common ancestor. Given sufficient epigenetic drift, genes likely have already sampled many different epigenetic configurations. If there is an optimal or near optimal epigenetic state, these cells will eventually replace lesser fit cells, especially in proliferative conditions such as normal epithelium or tissue culture (Fig 1A).

After epigenome optimization, replication errors still occur, but now most changes in functional genes decrease fitness, favoring cell loss from negative selection. In this phase, conservation can be used as a metric of gene functionality because drift preferentially accumulates in nonfunctional regions. More essential human genes are more likely to have reached an optimal epigenetic configuration from positive selection, and less likely to tolerate even minor alterations due to negative selection. Given sufficient time between measurements, cells can signal through conservation which genes are more likely to be essential to their survival. Next, we illustrate that DepMap essential human genes exhibit preferential conservation, and observations consistent with a mechanism of negative selection acting on drift.

### Functional genes exhibit preferential epigenetic conservation

DNA methylation was compared between two replicates of SW620 colorectal cancer (CRC) cell lines cultured in Spain and the USA, which are separated by many passages. (Fig 1B). Gene methylation was variable between the samples. To see if gene methylation differences are smaller for more functional genes, we used DepMap data [3] to identify "essential" genes that

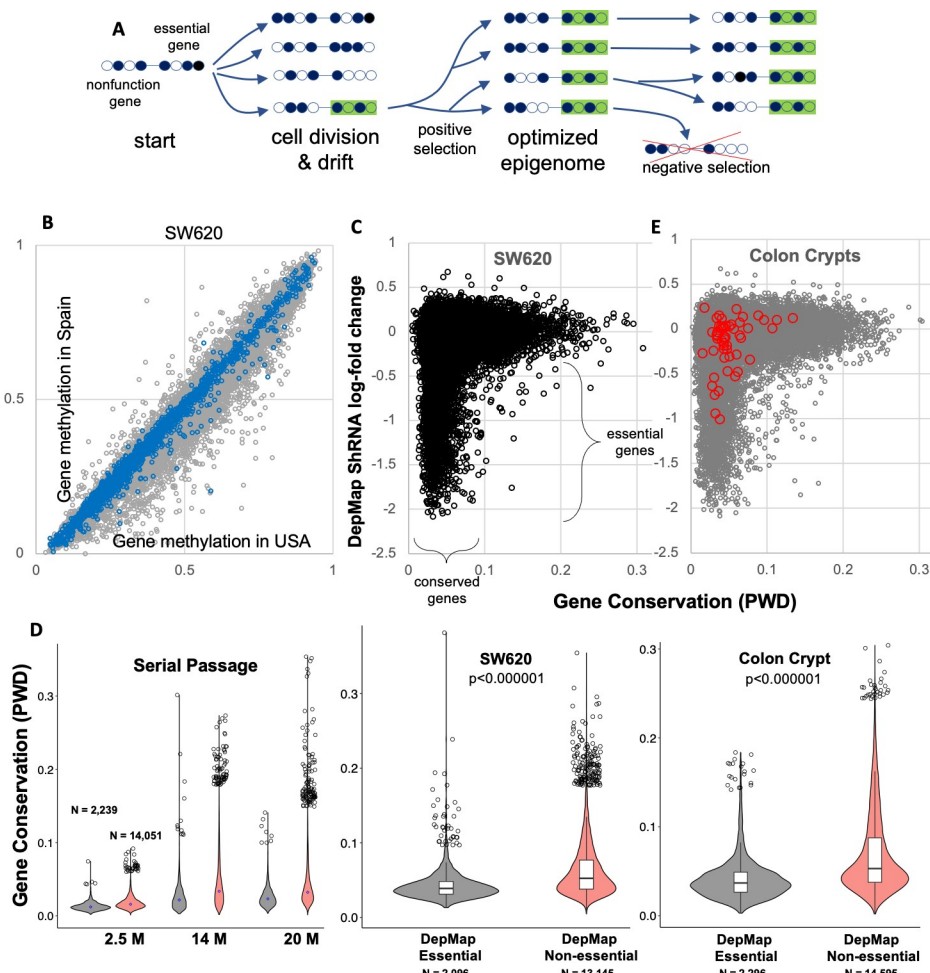

**Fig 1. DepMap essential genes exhibit preferential epigenetic conservation. A)** Diagram of how drift and selection cause preferential epigenetic conservation. Initially identical daughter cells acquire epigenetic differences from drift. Cells with more optimal epigenomes (green) will be more fit and dominate the population. Once an epigenome is optimized, negative selection will tend to conserve the favored epigenetic configuration because variants are lost. Drift still occurs but changes are more tolerated in less functional genes, leading to preferential conservation of the genes under selection. The more essential a gene is to cell survival, the greater the selection and conservation. Circles indicate CpG sites, with filled circles indicating DNA methylation. **B)** Gene DNA methylation varies between two samples of the SW620 CRC cell line cultured in the USA and Spain. Smaller differences in methylation are present for DepMap essential genes (blue dots). **C)** DepMap essential genes (shRNA log-fold decrease less than -0.4) showed significantly smaller gene PWDs than non-essential genes. **D)** Essential genes are significantly more conserved (lower PWDs) than non-essential genes (red symbols) in tissue culture and normal colon crypts. Essential and non-essential genes in colon crypts were significantly less conserved than after 20 months of serial passage of a clonal cell line (p <0.000001). **E)** Essential DepMap genes are also significantly more conserved in normal human colon. Conservation differences are greater between normal adult colon crypts than between cell lines samples, likely reflecting the many more divisions that occur with aging relative to tissue culture. Most curated genes (red circles) important to colon crypt biology by other experimental criteria [10] are not essential by DepMap gene disruptions, but show preferential conservation, with gene PWDs less than 0.1.

reduced proliferation to less than -0.4 after single gene disruptions in CRC cell lines (N = 2,296 genes). DepMap essential genes were significantly more conserved (lower CpG methylation pairwise differences (PWD)) than non-essential genes (Fig 1B–1D). We extended these observations to normal colon, where the DNA methylation of four individual colon crypts were compared within 8 individuals (Fig 1D & 1E). Like the replicate CRC cell lines, DepMap essential genes were also preferentially conserved. The "r-shaped" distribution

**Table 1. Sensitivity, specificity, Positive Predictive Value (PPV), and Negative Predictive Value (NPV) of PWD thresholds for DepMap essential gene identification.**

| Metric | PWD < 0.05 | PWD < 0.10 | PWD < 0.15 |
|---|---|---|---|
| Sensitivity | 1,183/2,234 (53.0%) | 2,149/2,234 (96.2%) | 2,219/2,234 (99.3%) |
| Specificity | 10,053/14,033 (71.6%) | 3,449/14,033 (24.6%) | 1,116/14,033 (8.0%) |
| PPV | 1,183/5,163 (22.9%) | 2,149/12,733(16.9%) | 2,219/15,136(14.7%) |
| NPV | 10,053/11,104 (90.5%) | 3,449/3,534 (97.6%) | 1,116/1,131(98.6%) |

indicates most essential genes are conserved whereas less essential genes can drift and accumulate many more epigenetic differences through time.

Many genes important in normal colon may not score as essential in transformed cells during tissue culture. To extend the list of functional genes, we curated from a recent review [10] a list of 55 genes (S1 Table) considered important in colon crypt biology (red circles in Fig 1E). Only 9 (16%) of the curated genes scored as essential in DepMap disruptions whereas 51 (93%) of the curated genes had PWDs less than 0.1 in colon crypts. Therefore, tissue conservation is a more sensitive metric of curated tissue gene functionality.

Although most essential genes were conserved, many other genes were also conserved. Table 1 illustrates the sensitivity and specificity of DepMap essential gene identification based on its PWD in normal colon crypts. With a PWD threshold of 0.05, sensitivity was 53.0% and specificity was 71.6%. With a PWD threshold of 0.1, sensitivity increased to 96.2% and specificity decreased to 24.6%. Therefore, although most DepMap essential genes are conserved, many other genes are conserved, which may reflect epigenetic stability or unannotated functional roles.

## Evidence of epigenetic drift

The method relies on random alterations and negative selection to identify more functional genes. To experimentally demonstrate epigenetic drift, we measured methylation in clonal cell lines started from a single cell and serially passaged in triplicate (Fig 2A). As expected, the epigenomes of daughter cells were similar after 2.5 months, and all genes were conserved. After 14 months, overall conservation was less, and there were greater differences between DepMap essential and non-essential genes, with the r-shaped distributions maintained at 20 months (Fig 2A). The progressive loss of conservation is consistent with random drift in all genes, with higher conservation maintained in more functional genes. Consistent with even great intervals since birth, PWDs between both DepMap essential and other genes were greater in adult colon than between the serial clonal tissue cultures (Fig 1D). Therefore, the DNA methylation of all genes exhibit evidence of drift, with preferential epigenetic conservation of DepMap essential genes both in vitro and in vivo, and in normal and neoplastic cells.

## Evidence of negative or purifying selection

Conservation can be maintained by negative selection after an epigenome has been optimized because variants with less optimal epigenomes proliferate less (Fig 1A). Normally it is difficult to observe negative selection because the process of generating and eliminating variants is slow. However, substantial epigenetic variation can be created after treatment with the demethylating drug 5-azacitidine (AZA), and the fates of these variants can be measured after the drug is removed and cells recover. With purifying selection, the epigenome after recovery should resemble the optimized epigenome before treatment.

We analyzed a dataset where methylation was globally disrupted by low dose (500 nM) AZA treatment in 14 CRC cell lines [11]. DNA methylation was compared between mock treated controls and treated cultures during the recovery period after AZA treatment. Conservation immediately decreased after AZA treatment for both DepMap essential and other genes, but there was a trend towards greater conservation back to the control configurations within 21 days (Fig 2B).

A longer time course [12] after low dose (300 nM) AZA treatment with a single CRC cell line (HCT116) also showed initial loss of conservation of DepMap essential and other genes followed by regression back towards the original epigenome, and r-shaped distributions (Fig 2C & 2D). In this experiment, cell proliferation decreased immediately after AZA treatment but returned to the baseline doubling time by 68 days, indicating that cells with disrupted epigenomes are less fit. This increase in conservation with time (Fig 2D) contrasts with the decrease in conservation due to drift after single cell cloning (Fig 2A).

These cell line observations indicate epigenomes are subject to drift counteracted by negative selection (Fig 1A). Epigenetic drift accumulates slowly in all genes over months and

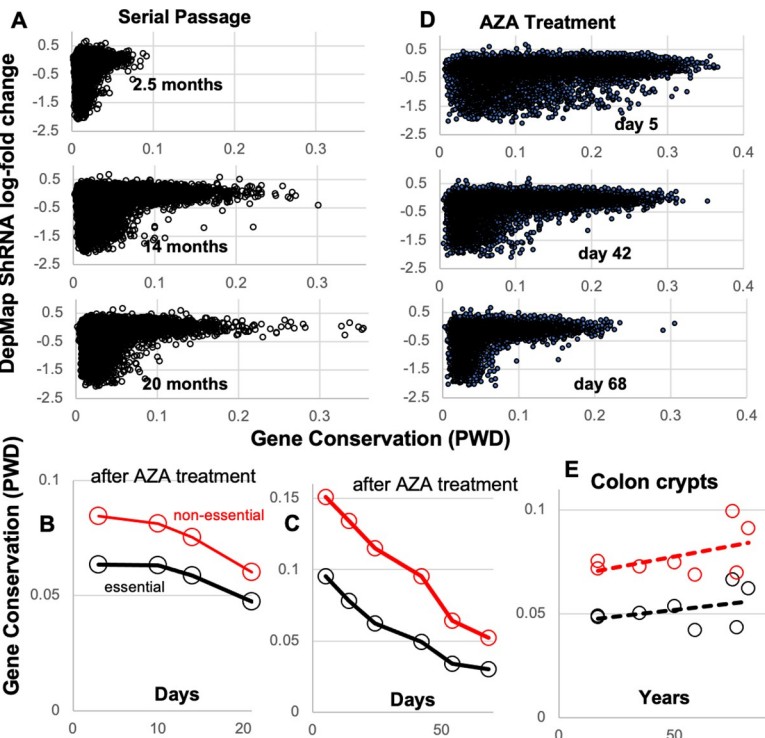

**Fig 2. Epigenetic drift and conservation. A)** Epigenetic drift occurs in all genes. Conservation progressively decreased between clonal cell lines started from a single colorectal cancer cell (HCT116) and passaged in triplicate. Differences between essential and non-essential genes are minimal after 2.5 months. Conservation decreases slightly for the essential genes but non-essential genes drift more by 14 months. The 14 and 20 month time points are similar (also see Fig 1D). **B)** PWDs between mock treated and AZA treated cultures (averages of 14 CRC cell lines). AZA treatment initially decreased conservation, but conservation progressively increased during the 21 days after DNA demethylation. The trend is marginally significant for the non-essential genes (p = 0.049) but not for DepMap essential genes (p = 0.11). **C)** PWDs between control and an AZA treated HCT116 CRC cell line. AZA also decreased conservation, and conservation progressively increased during the 68 days after DNA demethylation. The increase in conservation is significant for DepMap essential and non-essential genes (p<0.05). **D)** The increase in conservation with time after AZA treatment for the HCT116 CRC cell line contrasts with the decrease in conservation after single cell cloning (Fig 2A). **E)** Epigenetic conservation is stable with human aging in colon crypts. The trends for decreased conservation with aging are not significant.

purifying selection can act rapidly within weeks to eliminate cells with less optimal epigenomes, resulting in preferential conservation of more functional genes.

## Stable epigenetic conservation with aging

To further characterize conservation over a lifetime, we compared DNA methylation in colon crypts of different aged individuals (Fig 2E). Both DepMap essential and other genes show stable conservation, with a slight non-significant trend for less conservation with aging. Hence drift may continue to accumulate but preferential conservation is largely maintained over a lifetime. The conservation of all genes with aging may reflect that some non-DepMap essential genes have functional roles in normal crypts.

## Cell specific conservation

Genes that are more important for survival in specific cell types can be potentially identified by comparing their conservation in different tissues. We compared conservation within colon crypts with conservation within small intestinal (SI) crypts (4 crypts each from 4 individuals) and endometrial glands (4 glands each from 8 individuals). DepMap essential genes were generally conserved to the same degrees between tissues (Fig 3A). Genes with tissue specific conservation were relatively few between colon and SI crypts (S2 Table), and a Reactome pathway analysis [13] was uninformative (entities false discovery rates $> 10^{-3}$). More differences in tissue specific gene conservation were seen between the colon and endometrial glands, as might be expected with more divergent tissues. Again, a pathway analysis was uninformative. However, manual inspection revealed enrichment of differentially conserved homeobox genes (S2 Table). HOXA and other homeobox genes were more conserved in colon crypts whereas HOXB and other homeobox genes were more conserved in endometrial glands (Fig 3B). Homeobox genes are important master transcription factors that specify anterior-posterior cell identity during development [14–16], and differentially conserved homeobox genes may indicate tissue specific roles in adult epithelial renewal.

## Gene conservation signals drug sensitivity

The idea that conservation can identify which genes are more essential to cell survival can be further tested with cancer cell lines that differ in their drug sensitivities. A non-conserved gene may be a poor drug target because its cells can tolerate wide variations in its regulation. A gene

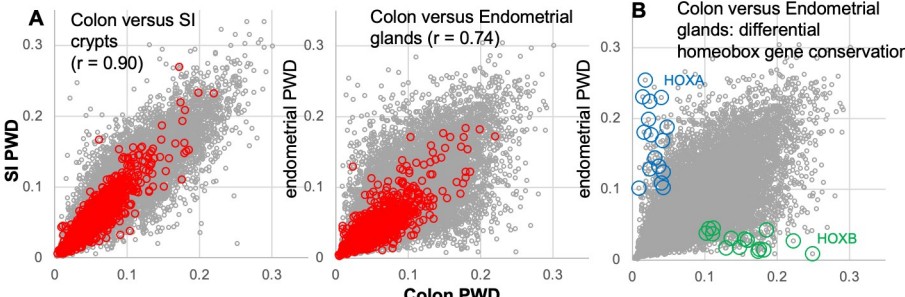

**Fig 3. Tissue specific gene conservation and conserved gene expression. A)** Conservation differences are greater between colon crypts and endometrial glands than between colon and SI crypts. This variation is less for DepMap essential genes (red circles). Gene exhibiting differential tissue conservation may be important for tissue specific cell survival. **B)** Homeobox genes are enriched in differentially conserved endometrial versus colon comparisons. HOXA genes (blue circles) are preferentially conserved in colon crypts whereas HOXB genes (green circles) are preferentially conserved in endometrial glands.

with higher conservation may be a better drug target because it signals that even small functional differences in that gene are more likely to reduce the fitness and survival of its cells. Hence, drugs acting on conserved genes may be more effective.

This idea was tested with NCI-60 cell lines that vary in their drug sensitivities [17] and gene conservations (Fig 1B). We examined the conservation at ten genes with respect to their targeted drug sensitivities (Table 2). These genes were chosen because they produce the molecular targets of the drugs (Genomics of Drug Sensitivity in Cancer) [18]. The cell line sensitivities of 12 of the 16 targeted drugs correlated with higher targeted gene conservation. Three drugs targeted against EGFR showed a dose response with respect to conservation (Fig 4A). Higher BCL2, ALK, and FLT3 conservation also correlated with greater targeted drug sensitivity (Fig 4B). Correlations were weak but drug sensitivity exhibited r-shaped distributions, where the most sensitive cell lines generally had more conserved target genes. A gene may be targetable when highly conserved, but the same gene rarely conferred drug sensitivity if not conserved in their cells.

## Discussion

Recent systematic CRISPR-Cas9 studies [3] have identified hundreds of essential genes defined by decreased proliferation in culture when the gene is disrupted [1–3]. Here we show that DepMap essential genes also show preferential epigenetic conservation in cell lines and in human tissues. DNA methylation can be easily measured, and measuring epigenetic conservation simply requires one additional measurement separated by time. Analogous to experimental gene disruptions, cells with epigenetic alterations that reduce survival will be lost during this interval, and therefore two measurements can indicate the extent of epigenetic variation tolerated by the survivors. We showed that the epigenomes of all genes drift in tissue culture with time. The preferential conservation of DepMap essential genes indicate that drift in these genes is less tolerated both in tissue culture and in normal human epithelium. Hence, the extent of epigenetic conservation can be used as a signal to infer or rank the relative importance of a gene to cell survival. Whereas epigenetic drift accumulates over months (Fig 2A), purifying selection

**Table 2. Correlation between cell line target gene conservation and targeted drug therapy sensitivities.**

| Drug | Target | Pearson | P value |
|---|---|---|---|
| MIDOSTAURIN | FLT3 | 0.326015 | 0.018 |
| TEMSIROLIMUS | MTOR | 0.136519 | 0.335 |
| CRIZOTINIB | ALK | 0.133021 | 0.357 |
| | MET | 0.038333 | 0.792 |
| ALECTINIB | ALK | -0.03998 | 0.778 |
| | MET | -0.02822 | 0.843 |
| IRINOTECAN | TOP1 | 0.290191 | 0.043 |
| TOPOTECAN | TOP1 | 0.09682 | 0.508 |
| CAMPTOTHECIN | TOP1 | 0.131144 | 0.369 |
| IBRUTINIB | BTK | -0.19832 | 0.172 |
| LAPATINIB | ERBB2 | 0.238359 | 0.099 |
| ERLOTINIB | EGFR | 0.214675 | 0.139 |
| CETUXIMAB, ERBITUX | EGFR | 0.253057 | 0.065 |
| LAPATINIB | EGFR | 0.196187 | 0.177 |
| GEFITINIB | EGFR | 0.047803 | 0.729 |
| VENETOCLAX | BCL2 | 0.218137 | 0.145 |
| OSIMERTINIB | EGFR | -0.18272 | 0.224 |
| DABRAFENIB | BRAF | -0.11179 | 0.416 |

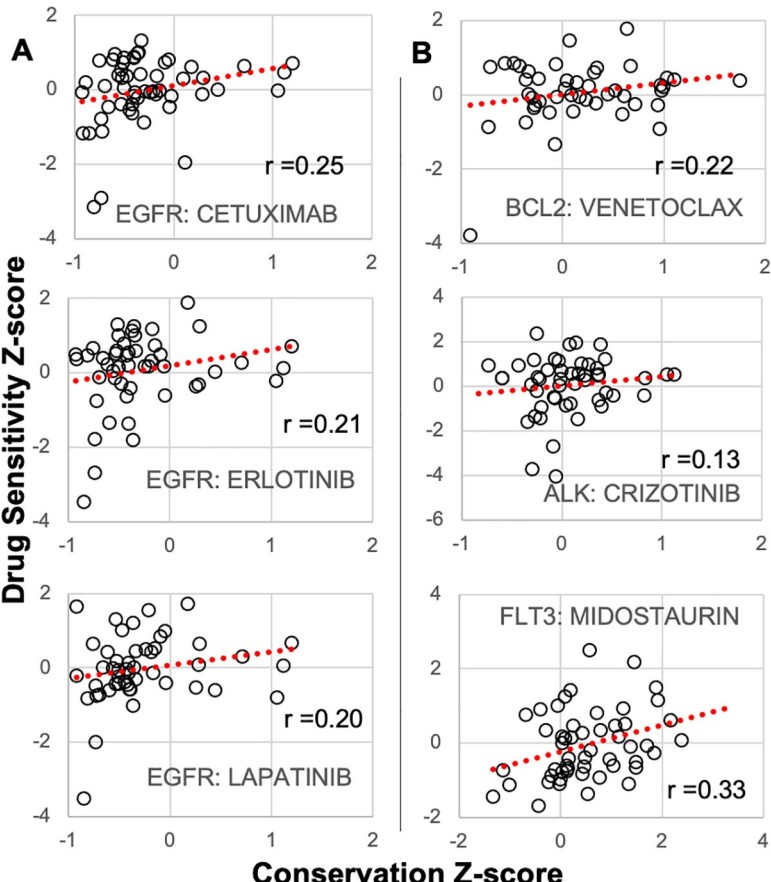

**Fig 4. Gene conservation signals drug vulnerabilities. A)** Targeted EGFR drug sensitivities (z-scores, negative values more drug sensitive) correlated with EGFR conservation (z-scores, negative values more conserved). Most drug sensitive cell lines showed preferential targeted gene conservation with r-shaped distributions. **B)** Conservation of BCL2 (venetoclax), ALK (crizontinib), and FLT3 (midostaurin) correlated with their cell line targeted therapy sensitivities.

can act within weeks to maintain the preferential epigenetic conservation of more functional genes (Fig 2D).

Essential DepMap genes are generally expressed at high levels [19], and cancer cell lines exhibit similar DepMap essential genes, consistent with their housekeeping functions [2]. Similarly, most essential DepMap genes were also conserved in colon crypts, SI crypts and endometrial glands. The sensitivity of conservation for essential genes was high, and genes with low DNA methylation conservation were rarely essential in DepMap studies. However, the specificity of epigenetic conservation to identify essential genes was low because many genes that did not score as essential by CRISPR-Cas9 disruptions were also conserved. Essentiality is cell type and context dependent, and most curated genes that are considered important to crypt biology by other experimental data [10] exhibited conservation in normal crypts but did not score as essential in cell line disruptions. Other conserved genes may have unknown or untested roles or have less epigenetic drift. Given the limitations of human functional tissue studies, differential gene conservation, as observed with homeobox family genes, may help unravel which genes are more essential in different tissues.

Methods to identify essential human genes are imperfect and different approaches are often complementary [1, 2]. Although gene function defined by epigenetic conservation is

inferential, conservation correlated well with more direct experimental evidence. DNA methylation has the advantage of being easily measured across the genome by widely used commercial microarrays with standardized bioinformatics [8, 9]. Hence, measurements are relatively easy to compare between different time intervals, tissues, and laboratories, allowing indirect orthogonal inferences of gene essentiality that are otherwise difficult to perform in normal human tissues and microenvironments. Much of the data analyzed here were extracted from publicly available data sets. A better understanding of which CpG sites correlate with regulation could improve the detection of genes under selection.

A method to infer which genes are more essential to cell survival would be useful to guide cancer therapy. Cancer cells may signal their vulnerabilities by conserving the DNA methylation of their more important genes, and genes involved with immune surveillance were preferential conserved during human CRC growth [7]. Cancer cell line drug sensitivities can be predicted by considering mutation, expression, and methylation parameters [17], but modeling often suffers from low interpretability [20]. As illustrated with a subset of targeted therapies and their gene targets, epigenetic conservation may offer another, easy to understand, predictor of drug sensitivity. This potential biomarker should improve with disease progression, such as relapse after standard therapies, because greater time between sampling should increase the epigenetic variation in less critical genes. Genes consistently conserved during progression are more likely to be critical to the survival of their tumor cells. Conservation as a metric of gene function has the advantage that its readout from drift and selection occurs in the natural unperturbed tissue microenvironment. Normal and neoplastic cells may signal through epigenetic conservation which genes are more likely important to their survival.

## Materials and methods

### Experimental design

The objective of this study was to determine if the degree of epigenetic conservation of a gene correlates with its essentiality. Epigenetic conservation requires two measurements of the same cell population separated by sufficient amounts of time for drift to accumulate. Conservation is measured as a pairwise distance between the methylation (beta-values) at a CpG site and varies between 0 and 1. The conservation of a gene is the average pairwise distance of all CpG sites associated with that gene.

For a particular CpG with $k$ DNA methylation proportion measurements, let $A_i$ denote the $i^{th}$ DNA Methylation proportion measurement. The pairwise distance measure (PWD) of conservation can be computed through the following:

$$\frac{1}{\binom{k}{2}} \sum_{i=1}^{k-1} \sum_{j=i+1}^{k} |A_i - A_j|$$

where $\binom{k}{2} = \frac{k!}{2!(k-2)!}$ denotes the number of unique sample pairs for comparison.

Gene essentiality is taken from DepMap data [3] of single CRISPR-Cas9 gene disruptions and were downloaded from the DepMap website (https://depmap.org/portal/Public20Q4, Archilles_gene_effect.csv). We defined a gene as essential if disruption led to its shRNA log-fold change of less than -0.4. We used the average shRNA log-fold change from the 9 available CRC cell lines for analysis.

### Materials

The human samples were from excess tissues taken during routine clinical care at the USC Medical Center, with approval by Office for the Protection of Research Subjects at the

University of Southern California (HS-18-00043). Specific consent from patients or guardians was waived by the Office for the Protection of Research Subjects at the University of Southern California because the samples were excess tissues obtained in the course of routine clinical care. The data were analyzed anonymously. Single colon crypts (N = 32), small intestinal (SI) crypts (N = 16), and endometrial glands (N = 32) were isolated from excess normal colon (N = 8 individuals), SI (N = 4 individuals), or uteri (N = 8 individuals) using an EDTA shake-out procedure [21, 22]. The glands or crypts were greater than 90% epithelial cells. The serial clonal cell lines were started from a single CRC cell (HCT116) as previously reported [7].

DNA methylation array data in publicly available databases (beta values were used as provided) were obtained for NCI60 cell lines (GSE79185 from the National Cancer Institute, USA, and GSE49143 from Barcelona, Spain; CRC cell lines COLO205, HCC2998, HCT15, HCT116, HT29, KM12, SW620), and AZA treated CRC cells lines (GSE57342 and GSE51815). Cell line drug sensitivity data was also downloaded from the DepMap website (Sanger GDSC1 and GDSC2, sanger-dose-response.csv).

## Statistical analysis

Statistical Analyses were performed in Excel and R. Significant differences in central tendency were determined with Welch's unequal variances t-test. Pearson correlation analysis was used to indicate trend of data. Significance level for all tests was set to $P = 0.05$.

## Supporting information

**S1 Table. List of essential intestinal crypt genes.** Curated from: Nat Rev Gastroenterol. Hepatol. 2019;16:19–34.
(XLSX)

**S2 Table. List of genes preferentially conserved in colon crypts, SI crypts and endometrial glands.** More conserved is defined as gene PWD<0.5 in the conserved tissue and PWD>.1 in the other tissue. Red lettering identifies HOX and homeobox genes.
(XLSX)

## Author Contributions

**Conceptualization:** Darryl Shibata.

**Data curation:** Daniel Rud, Kimberly Siegmund, Darryl Shibata.

**Formal analysis:** Daniel Rud, Kimberly Siegmund, Darryl Shibata.

**Funding acquisition:** Darryl Shibata.

**Investigation:** Daniel Rud.

**Methodology:** Daniel Rud, Paul Marjoram, Kimberly Siegmund, Darryl Shibata.

**Software:** Daniel Rud.

**Supervision:** Kimberly Siegmund, Darryl Shibata.

**Writing – original draft:** Daniel Rud, Darryl Shibata.

**Writing – review & editing:** Daniel Rud, Paul Marjoram, Kimberly Siegmund, Darryl Shibata.

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
