## [Decision Letter · Decision Letter 0]

14 Jul 2021

PONE-D-21-17428

Functional Human Genes Typically Exhibit Epigenetic Conservation

PLOS ONE

Dear Dr. Shibata,

Thank you for submitting your manuscript to PLOS ONE. After careful consideration, we feel that it has merit but does not fully meet PLOS ONE’s publication criteria as it currently stands. Therefore, we invite you to submit a revised version of the manuscript that addresses the points raised during the review process.

Please submit your revised manuscript within 30 days.  If you will need more time than this to complete your revisions, please reply to this message or contact the journal office at plosone@plos.org. Please include the following items when submitting your revised manuscript:

We look forward to receiving your revised manuscript.

Kind regards,

Anilkumar Gopalakrishnapillai

Academic Editor

PLOS ONE

Journal Requirements:

Reviewers' comments:

Reviewer's Responses to Questions

**Comments to the Author**

1. Is the manuscript technically sound, and do the data support the conclusions?

Reviewer #1: Partly

Reviewer #2: Yes

2. Has the statistical analysis been performed appropriately and rigorously? 

Reviewer #1: I Don't Know

Reviewer #2: Yes

3. Have the authors made all data underlying the findings in their manuscript fully available?

Reviewer #1: Yes

Reviewer #2: Yes

4. Is the manuscript presented in an intelligible fashion and written in standard English?

Reviewer #1: Yes

Reviewer #2: Yes

5. Review Comments to the Author

Reviewer #1: The topic is interesting, study approach is original and results of this study have potential translational meaning.

I have some points for improvement of the manuscript.

1. Number of bioethics committee aprroval is missing.

2. Page 2, line 21 – what does it mean „impractical” in this particular context (experimental approaches in human tissues based on CRISPR method). Please precise whether it concerns in vivo studies or tissue cultures. I would also use word „unethical” instead of „impractical” in the first case...

3. Page 5, line 76 – should it be NCI-60 mentioned here? Or rather SW620?

4. Table 2 - p values missing. I am not sure whether calculating average correlation coefficient makes any sense here, since some correlations are positive and some are negative.

Reviewer #2: I ll try to be brief.The paper is sound and I have not big concerns about experimental design or data analysis

However I think the authors should better explain the rationale behind the drug selection they have used

6. PLOS authors have the option to publish the peer review history of their article (what does this mean?). If published, this will include your full peer review and any attached files.

Reviewer #1: No

Reviewer #2: No

---

## [Author Response · Author response to Decision Letter 0]

21 Jul 2021

We thank the two reviewers for their efforts and comments.

We address their concerns as follows:

Reviewer #1

1) The missing bioethics approval number has been added to the Methods

2) We have change “impractical” to “unethical” as requested

3) Page 5, line 76: SW620 is correct. We have made the requested change.

4) Table 2: P values have been added as requested. We agree that calculating an average correlation coefficient does not make much sense and we have deleted the “average” line in Table 2. The positive and negative correlations are further described in the text on page 9.

Reviewer #2

 We have added more information on the choice of genes and their targeted therapies in the Results on page 9. The products of these genes are specifically targeted by these drugs, which allows for relatively simple single gene/drug comparisons for the NCI-60 cell lines. This information and an additional reference to where this information was obtained has been added.

---

## [Editor Report · Decision Letter 1]

9 Aug 2021

Functional Human Genes Typically Exhibit Epigenetic Conservation

PONE-D-21-17428R1

Dear Dr. Shibata,

We’re pleased to inform you that your manuscript has been judged scientifically suitable for publication and will be formally accepted for publication once it meets all outstanding technical requirements.

Kind regards,

Anilkumar Gopalakrishnapillai

Academic Editor

PLOS ONE

Additional Editor Comments (optional):

Thank you for addressing all the concerns of the reviewers and we have now accepted this manuscript for the publication. Congratulations.
---

## [Editor Report · Acceptance letter]

4 Sep 2021

PONE-D-21-17428R1 

Functional Human Genes Typically Exhibit Epigenetic Conservation 

Dear Dr. Shibata:

I'm pleased to inform you that your manuscript has been deemed suitable for publication in PLOS ONE. Congratulations! Your manuscript is now with our production department. 

Kind regards, 

on behalf of

Dr. Anilkumar Gopalakrishnapillai 

Academic Editor

PLOS ONE